# In Vitro Evaluation of Antiproliferative Properties of Novel Organotin(IV) Carboxylate Compounds with Propanoic Acid Derivatives on a Panel of Human Cancer Cell Lines

**DOI:** 10.3390/molecules26113199

**Published:** 2021-05-27

**Authors:** Nebojša Đ. Pantelić, Bojan Božić, Bojana B. Zmejkovski, Nebojša R. Banjac, Biljana Dojčinović, Ludger A. Wessjohann, Goran N. Kaluđerović

**Affiliations:** 1Department of Engineering and Natural Sciences, University of Applied Sciences Merseburg, Eberhard-Leibnitz-Straße 2, DE-06217 Merseburg, Germany; 2Department of Chemistry and Biochemistry, Faculty of Agriculture, University of Belgrade, Nemanjina 6, 11080 Zemun, Belgrade, Serbia; nbanjac@agrif.bg.ac.rs; 3Department of Bioorganic Chemistry, Leibniz Institute of Plant Biochemistry, Weinberg 3, 06120 Halle (Saale), Germany; Ludger.Wessjohann@ipb-halle.de; 4Faculty of Biology, Institute of Physiology and Biochemistry “Ivan Djaja”, University of Belgrade, Studentski trg 16, 11000 Belgrade, Serbia; bbozic@bio.bg.ac.rs; 5Department of Chemistry, Institute of Chemistry, Technology and Metallurgy, University of Belgrade, Studentski trg 14, 11000 Belgrade, Serbia; bojana.zmejkovski@ihtm.bg.ac.rs (B.B.Z.); bmatic@chem.bg.ac.rs (B.D.)

**Keywords:** triphenyltin(IV), breast cancer, cytotoxicity, apoptosis, ICP-MS

## Abstract

The synthesis of novel triphenyltin(IV) compounds, Ph_3_SnL*n* (*n* = 1–3), with oxaprozin (3-(4,5-diphenyloxazol-2-yl)propanoic acid), **HL1**, and the new propanoic acid derivatives 3-(4,5-bis(4-methoxylphenyl)oxazol-2-yl)propanoic acid, **HL2**, and 3-(2,5-dioxo-4,4-diphenylimidazolidin-1-yl)propanoic acid, **HL3**, has been performed. The ligands represent commercial drugs or their derivatives and the tin complexes have been characterized by standard analytical methods. The in vitro antiproliferative activity of both ligands and organotin(IV) compounds has been evaluated on the following tumour cell lines: human prostate cancer (PC-3), human colorectal adenocarcinoma (HT-29), breast cancer (MCF-7), and hepatocellular cancer (HepG2), as well as on normal mouse embryonic fibroblast cells (NIH3T3) with the aid of MTT (3-(4,5-dimethylthiazol-2-yl)-2,5-12 diphenyltetrazolium bromide) and CV (crystal violet) assays. Contrary to the inactive ligand precursors, all organotin(IV) carboxylates showed very good activity with IC_50_ values ranging from 0.100 to 0.758 µM. According to the CV assay (IC_50_ = 0.218 ± 0.025 µM), complex **Ph_3_SnL1** demonstrated the highest cytotoxicity against the caspase 3 deficient MCF-7 cell line. Inductively coupled plasma mass spectrometry (ICP-MS) analysis indicated a two-fold lower concentration of tin in MCF-7 cells in comparison to platinum. To investigate the mechanism of action of the compound **Ph_3_SnL1** on MCF-7 cells, morphological, autophagy and cell cycle analysis, as well as the activation of caspase and ROS/RNS and NO production, has been performed. Results suggest that **Ph_3_SnL1** induces caspase-independent apoptosis in MCF-7 cells.

## 1. Introduction

Carcinogenic diseases represent a spectrum of related disorders, characterized by uncontrolled cell division in individual organs, with a tendency to invade other parts of the body. Viruses appear as causative agents of about 15 percent of all human cancers worldwide [1]. Additionally, a large fraction of cancerous diseases arise spontaneously, while a significant share is caused by exposure to ionising radiation [2]. Food enriched with mycotoxins, different hormones and other compounds represent another important source of cancer-causing agents [3,4].

The discovery of cisplatin paved the way for the treatment of oncological diseases with metal-based compounds. Unfortunately, the use of cisplatin and other platinum derivatives for the treatment of malignant tumours causes a series of side effects such as nausea, vomiting, neurotoxicity, nephrotoxicity, and ototoxicity [5,6]. To reduce these side effects of metal-based cytotoxins, the focus of novel investigations shifted to non-platinum candidates [7]. Organotin compounds exhibit an excellent potential for varying the organic moieties of the donor ligand linked to the metal, which enables fine-tuning of the physico-chemical properties of such novel coordination compounds. Consequently, a variety of diorganotin and triorganotin(IV) complexes with significant in vitro antiproliferative activity against solid and hematologic cancers have been synthesised [8]. More recently, the tin complexes ((carboxylato)triphenyltin(IV) and tetraorganotin(IV)) have been found to display even higher cytotoxic activity than cisplatin [9,10]. Compared to platinum drugs, organotin(IV) compounds benefit from better water solubility, lower toxicity and reduced side effects [11]. Noteworthily, these types of complexes display a strong apoptotic effect on tumour cells, via p53 tumour suppression [11]. Organotin(IV) carboxylates, compared to organotin(IV) compounds with thiolato and ditiocarbamato ligands, usually display higher cytotoxic activity against human tumour cell lines [12].

Investigation of the triphenyltin(IV) complex with 2,5-dimethyl-3-furoic acid as a ligand showed high activity, especially against K562 and Fem-x cell lines, with IC_50_ values 30 to 112 times lower than cisplatin. Additionally, a relatively high selectivity vs. non-tumour immortalized cells was demonstrated for this complex [13]. Other triphenyltin(IV) carboxylates also show very high activity against different cell lines, displaying even greater cytotoxic activity compared to cisplatin [14]. Recent studies of the triphenyltin(IV) compounds of 1-(4-carboxyphenyl)-3-ethyl-3-methysuccinimide demonstrated their significant activity against the HeLa cells [15]. Furthermore, organotin(IV) complexes of 2-(5-arylidene-2,4-dioxothiazolidin-3-yl)propanoic acid yielded excellent improvements in antiproliferative drug development against PC-3 cells [16].

In line with our continuing efforts for developing novel organotin compounds as potential antiproliferative agents, we herein report the synthesis of novel organotin(IV) complexes with the commercially available drug Oxaprozin as well as two novel propanoic acid derivatives. Oxaprozin represents a non-steroid anti-inflammatory drug (NSAID), whose metal complexes previously displayed antiproliferative properties. All compounds, ligand precursors and complexes synthesised in this study have been characterized by standard analytical methods, and the corresponding in vitro anticancer activities were determined on four tumour cell lines: human prostate cancer (PC-3), human colorectal adenocarcinoma (HT-29), breast cancer (MCF-7), and hepatocellular cancer (HepG2), with the aid of MTT and CV assays. The mode of action of the novel organotin(IV) complex **Ph_3_SnL1** on MCF-7 cell lines has been further evaluated via morphological, autophagy and cell cycle analysis, showing the activation of caspase as well as ROS/RNS and NO production. In addition, the accumulations of tin and platinum in MCF-7 cells have been compared.

## 2. Results and Discussion

### 2.1. Synthesis and Characterization

In the reaction of ligand precursors (**HL1**, **HL2**, or **HL3**) which were deprotonated with LiOH and reacted with equimolar amounts of Ph_3_SnCl, the expected compounds **Ph_3_SnL1**, **Ph_3_SnL2** or **Ph_3_SnL3** were obtained as white solids in good yields (Scheme 1a–c). The complexes were soluble in dimethyl-sulfoxide, chloroform, dichloromethane, and acetonitrile. They were characterized via elemental microanalysis, IR and NMR (^1^H, ^13^C, ^119^Sn) spectroscopy and mass spectrometry. The numbering of atoms in Scheme 1a–c was only used for NMR data assignation.

The elemental analysis corresponds to the predicted formulae. Asymmetric ν(–CH_3_/–CH_2_/–CH) vibrations of moderate intensities were found in the range of 3200–2800, corresponding to unsaturated CH bonds (above 3000) and saturated CH bonds (below 3000) cm^−1^. Further, in the known aromatic vibration region (2000–1600 cm^−1^) of complexes compared to ligand precursors, a pattern change was observed which indicates the additional presence of aromatic moieties in the novel compounds. Strong ν(C=O) and ν(C–O) absorption stretching bands are present in all three complexes, at 1648, 1432 cm^−1^ (**Ph_3_SnL1**); 1713, 1452 cm^−1^ (**Ph_3_SnL2**) and 1710, 1419 cm^−1^ (**Ph_3_SnL3**). Noteworthy, in the spectra of ligand precursors, these bands were found at 1720, 1443 cm^−1^ (**HL1**), 1708, 1436 cm^−1^ (**HL2**), and 1701, 1419 (**HL3**) cm^−1^. This suggests a monodentate coordination of the carboxylate oxygen atom as carbon–oxygen bond lengths are affected, becoming unequal. The C=O bond vibrates at higher wavelength values which is most visible in the case of **HL1/Ph_3_SnL1** [17]. Differences between asymmetric and symmetric vibrations were more than 200 cm^−1^, confirming a monodentate coordination to the tin centre [18]. Additionally, medium ν(C=O) bands at 1777 (**HL3**) and 1770 (**Ph_3_SnL3**) cm^−1^ confirm the presence of hydantoin rings. Moreover, a weak band at 454, 447 and 449 cm^−1^, observed for all three compounds, respectively, is due to Sn–O vibrations, strongly suggesting the formation of organotin(IV) carboxylates [15,16]. The ESI-MS of all synthesised organotin(IV) compounds has been recorded in positive ion mode and in all cases the [M + H]^+^ mass peak was found.

In ^1^H NMR spectra, for all compounds the characteristic pattern for the –CH_2_CH_2_– group from the propanoic acid moiety can be observed at around 2.8–3.9 ppm. The chemical shifts at 3.76, 3.77 (**HL2**) and 3.83, 3.87 ppm (**Ph_3_SnL2**) belong to metoxyl hydrogen atoms. As expected, aromatic protons resonated between 6 and 8 ppm. Coupling of hydrogen atoms with the tin centre can be observed as satellite signals in all organotin compounds, with *ortho* hydrogen atoms near the positions given in the data (experimental section), but due to overlap with resonances of hydrogen atoms from the five phenyl moieties this cannot be easily distinguished for characterization. Ligand precursors have less shielded hydrogen atoms (between 11–12 ppm) belonging to their carboxylate groups. Expectedly, this resonance is not present in the organotin compounds, confirming coordination by the carboxylate group to tin(IV).

^13^C NMR spectra showed all expected signals for these types of compounds. Carbon atoms *C*^2^ and *C*^3^ from –CH_2_CH_2_– groups resonated in both ligand precursors and organotin(IV) compounds at 23–38 ppm. Metoxyl carbon atoms in **HL2**/**Ph_3_SnL2** resonated at 55.5 ppm. In the area 110–160 ppm, all resonances of aromatic carbon atoms can be found, nevertheless, overlapping was observed. Chemical shifts of two carbonyl carbon atoms from the hydantoin rings in **HL3**/**Ph_3_SnL3** can be allocated at around 156 and 173 ppm. The carbon atoms from the carboxylate moieties in organotin(IV) compounds as well as the ligand precursors had the highest ppm values, as presumed. Furthermore, ^119^Sn NMR spectra showed a singlet at −96.1, −95.0 and −89.3 ppm for Ph_3_SnL1, Ph_3_SnL2, and Ph_3_SnL3, respectively, suggesting the tetrahedral geometry of triphenyltin(IV) compounds [19].

In order to investigate the stability of the complexes, time-dependent ^13^C NMR spectroscopy was performed. The recorded ^13^C NMR spectra were compared immediately after dissolving the investigated compounds, and again after 2 and 24 h in CDCl_3_. The results show that at the examined time points there were no changes in the chemical shift resonances of carbon atoms, nor the appearance of new signals that can be attributed to decomposition products in the applied medium.

### 2.2. Cytotoxic Activity

The in vitro anticancer activity of ligand precursors, **HL1**–**HL3** and its organotin(IV) compounds was tested in a panel of four tumour cell lines: PC3 (prostate cancer cells), HT-29 (colon cancer cells), MCF-7 (breast cancer cells), and HepG2 (hepatic cancer cells), as well as on normal NIH3T3 cells (mouse embryonic fibroblast) using MTT and CV assays after 48 h of incubation. Results are presented in Table 1.

All investigated organotin(IV) compounds demonstrated outstanding antiproliferative activity versus all tested tumour cell lines with IC_50_ values ranging from 0.100–0.785 µM. Generally, compounds **Ph_3_SnL1** and **Ph_3_SnL3** exhibited a significantly higher cytotoxicity against all tumour cells compared to **Ph_3_SnL2**. Compound **Ph_3_SnL3** showed the strongest inhibitory effect on cell survival in HepG2 hepatic tumour cells, while compound **Ph_3_SnL1** was found to be the most active against MCF-7 breast cells (Table 1). Unfortunately, novel organotin(IV) compounds show high toxicity against normal NIH3T3 cells. However, obtained IC_50_ values are significantly higher toward tumour cell lines, thus, investigated organotin(IV) compounds showed 3 to 12 times higher ca. selectivity (CV assay) towards tumour cells. On the other hand, the ligand precursors **HL1**–**HL3** did not show any influences on cell proliferation at the applied concentrations.

Additionally, it should be noted that all analysed organotin(IV) compounds were several times more active compared to cisplatin, a clinically used drug (Table 1), especially compounds **Ph_3_SnL1** and **Ph_3_SnL3**, which displayed up to 120-fold higher activity against MCF-7. Despite the high toxicity, the selectivity between normal and cancer cells of novel compounds is higher or comparable to cisplatin.

Moreover, the triphenyltin(IV) compounds expressed significantly higher antiproliferative activity compared to Mn(II), Co(II), Ni(II), Cu(II) and Zn(II) complexes with oxaprozin as a ligand [20]. According to the results obtained by MTT assay, it was observed that compound **Ph_3_SnL3** had its highest activity against HepG2 cells, while compound **Ph_3_SnL1** shows the lowest IC_50_ value versus MCF-7 in the CV assay (Table 1). A discrepancy in MTT and CV assays should be noted, possibly pointing out that investigated organotin(IV) compounds affect cell metabolism pathways. Such effects have already been described in literature, e.g., as being caused by 2-oxoheptyl ITC [21]. To examine the mode of action, the compound **Ph_3_SnL1** and the MCF-7 tumour cell line were selected for further investigation.

### 2.3. Drug Uptake

The cellular uptake features of metal-based cytotoxic drugs strongly impact the underlying anticancer performance [22]. Intracellular tin and platinum uptake in MCF-7 cells has been investigated with the aid of inductively coupled plasma mass spectrometry (ICP-MS) analysis. The treatment of cells with **Ph_3_SnL1** and cisplatin for 24 h in concentrations corresponding to their IC_50_ values, as obtained by the CV assay, was performed.

The results (Figure 1) point to significantly different metal uptake, with emphasis on the approximately twofold higher platinum concentration in MCF-7 cells compared to tin. Interestingly, the results indicate that a minor amount of tin causes significantly more potent cytotoxic effect against MCF-7 cells, compared to the clinically used anticancer drug cisplatin. Considering the uptake, it should be emphasized that half the concentration of tin (compared to cisplatin) in MCF-7 cells results in up to 70 times more powerful antiproliferative activity.

### 2.4. Morphological Study

The results of morphological changes of MCF-7 cells obtained by AO and DAPI staining following treatment with **Ph_3_SnL1** and cisplatin in concentrations corresponding to their IC_50_ and 2 × IC_50_ values after 24 h of incubation, are presented in Figure 2. It can be observed that the cell treatment triggers morphological changes in MCF-7, indicating the possible induction of apoptosis, as identified by the permeable dye acridine orange. Compared to untreated cells, notable morphological changes are observed in cells treated by **Ph_3_SnL1** at both concentrations applied (IC_50_ or 2 × IC_50_). The level of nucleus condensation is considerably higher in treated compared to untreated control cells. Furthermore, the results obtained by DAPI staining confirmed nuclear condensation, suggesting apoptotic morphological changes in treated MCF-7 cells with **Ph_3_SnL1** (Figure 2). The control cells remained intact and uniform.

### 2.5. Flow Cytometry Analysis

The activation of caspase is closely linked to cell apoptosis, the controlled cell death, via the mitochondrial pathway or cell death receptor [23,24]. To evaluate whether caspases get activated to trigger apoptosis, MCF-7 cells were treated for 48 h with **Ph_3_SnL1**, and cisplatin as a reference substance, in the concentrations corresponding to their IC_50_ values, stained with apostat, and analysed using flow cytometry. As shown in Figure 3a, **Ph_3_SnL1** did not induce the activation of caspases, moreover, **Ph_3_SnL1** induced a down-regulation of caspase expression. Due to this, it can be concluded that cell death is accomplished through caspase-independent apoptosis. Moreover, as MCF-7 cell line is caspase-3 deficient [25,26], this was an anticipated result. Besides, the accomplishment of apoptosis induced with **Ph_3_SnL1** in MCF-7 cells (confirmed with AO and DAPI assays) was not disturbed as this type of cell death may be activated through other death signals not involving caspase activation [27].

The literature data show that autophagy in cancer cells plays a dual role in cell survival and cell death [28,29]. To investigate whether the test compounds could induce autophagy, MCF-7 cells were stained with AO after treatment at IC_50_ concentrations of **Ph_3_SnL1** and cisplatin for 48 h. The results obtained by flow cytometry analysis show that the autophagy process is not triggered by **Ph_3_SnL1**. On the contrary, cisplatin induces an autophagic response in MCF-7 cells at the applied concentration (Figure 3b).

The mode of action was further investigated by analysing the effects of **Ph_3_SnL1** and cisplatin on the cell cycle phase distribution of MCF-7 cells. This was evaluated by cytofluorimetric analysis using DAPI staining. The obtained results show that the incubation with **Ph_3_SnL1** for 48 h does not lead to any significant changes in the cell cycle distribution (Figure 4). The analogous treatment with cisplatin also did not lead to any observable changes in the population of cells in the sub-G1 phase, while a considerable decrease was noted for the G1/G0 phase, which was coupled with an increase in the cell population in S and G2/M phases (Figure 4). The exhibited results suggest that the investigated organotin(IV) compound, **Ph_3_SnL1**, has a mode of action different to cisplatin. While cisplatin acts on DNA (among other effects), the lipophilic cationic nature of triphenyltin(IV) may guide it preferentially to mitochondria to intervene with cells’ energy metabolism.

Reactive oxygen (ROS) and nitrogen species (RNS) are natural by-products of the aerobic metabolism in cells [30,31]. At physiological concentrations, they play a significant role in regulating numerous cellular processes. On the other hand, it is well-known that the reactive oxygen (O_2_^•−^, H_2_O_2_, OH^•−^) and nitrogen species (NO) are highly reactive molecules that easily damage cells by interacting with biomolecules such as proteins, lipids, carbohydrates, and nucleic acids, thus triggering different pathological conditions (inflammation, cardiovascular disorders, neurodegenerative diseases, and the formation of cancer cells) [32,33,34,35]. The influence of **Ph_3_SnL1** and cisplatin on the production of ROS/RNS species in MCF-7 cells was evaluated by flow cytometry treatment of the cells with tested compounds in IC_50_ concentrations for 48 h. The obtained results demonstrate that **Ph_3_SnL1** does not affect the production of ROS and RNS within MCF-7 cells, while cisplatin elevates the production of both reactive oxygen and nitrogen species in cancer cells (Figure 5a,b).

## 3. Materials and Methods

### 3.1. Measurements

Elemental analyses were performed on an Elemental Vario EL III microanalyzer. A Nicolet 6700 FT–IR spectrometer (THERMO Scientific, Waltham, MA, USA) and ATR technique were used for recording mid-infrared spectra (4000–400 cm^−1^). ^1^H and ^13^C NMR spectra were recorded on a Bruker Avance III 500 spectrometer (Bruker, Billerica, MA, USA). ^119^Sn NMR spectra were recorded on a Bruker Avance DRX 400 spectrometer. Biological investigations were performed on a microplate reader (Spectramax, Molecular Devices USA, Silicon Valley, CA, USA), a laminar flow cabinet (Herasafe KS, Thermofischer Scientific, Waltham, MA, USA) and FACSAria III, BD Biosciences, DB Biosciences, (Basel, Switzerland). Mass spectrometry was performed using API-3200 triple quadrupole MS (AB Sciex) Agilent 1200 with autosampler (Applied Biosystem, Framingham, MA, USA) with electrospray positive ionisation mode. Regents and solvents were purchased from Merck (Belgrade, Serbia) and used without further purification.

### 3.2. Synthesis of Ligands Precursors

The ligands, 3-(4,5-diphenyloxazol-2-yl)propanoate, **L1**, 3-(4,5-bis(4-methoxylphenyl)oxazol-2-yl)propanoate, **L2**, and 3-(2,5-dioxo-4,4-diphenylimidazolidin-1-yl)propanoate, **L3**, were synthesised by the neutralisation of previously prepared propanoic acid derivatives (**HL1**, **HL2** and **HL3**). **HL1**, the commercially available drug oxaprozin, was synthesised according to a previously published method and its chemical characterization (FT-IR, ^1^H-NMR and ^13^C-NMR spectra) fully matched previous spectra [20]. **HL2**, the *para*-methoxyl derivative of Oxaprozin, was been synthesised according to a previously published method [20] whereby the starting reagent 2-hydroxyl-1,2-bis(4-methoxylphenyl)ethanone (*para*-methoxyl derivative of benzoine) was used for **HL2**, instead 2-hydroxyl-1,2-diphenylethanone (benzoine) for **HL1**. Further, **HL3** is a propanoic acid derivative of the commercially available drug Phenytoin and was synthesised according to a previously published method [36].

3-(4,5-*bis*(4-Methoxylphenyl)oxazol-2-yl)propanoic acid, **HL2**: White solid; **^1^H NMR** (500 MHz, DMSO-*d_6_*): *δ* 2.77 (t, 2H, ^3^*J*_(H,H*)*_ = 7.5 Hz, C^2^H), 3.22 (t, 2H, ^3^*J*_(H,H)_ = 7.5 Hz, C^3^H), 3.76 (s, 3H, CH_3_O–Ph), 3.77 (s, 3H, CH_3_O–Ph), 6.90–7.20 (m, 4H, C^8^H, C^10^H,C^15^H, C^17^H), 7.30–7.60 (m, 2H, C^7^H, C^11^H*)*, 7.80–8.00 (m, 2H, C^14^H, C^18^H), 10.86 (s, 1H, COOH) ppm.^**13**^**C NMR** (125 MHz, DMSO-*d_6_*): *δ* 23.4 (C^3^), 30.4 (C^2^), 55.5 (*C*H_3_O–Ph), 114.7 (C^8^, C^10^, C^15^, C^17^), 125.6 (C^13^), 128.0 (C^14^, C^18^), 129.0 (C^7^, C^11^), 132.0 (C^6^), 133.9 (C^5^), 144.2 (C^12^), 159.8 (C^4^), 162.4 (C^9^, C^16^), 173.6 (C^1^) ppm. Selected **FT-IR** data (KBr) cm^−1^: 3006 [ν(CH)], 2928 [ν(CH)], 2638 [ν(CH2)], 1708 [ν(C=O)], 1586 [ν(CC)], 1517 [ν(CC)], 1436 [δ(CO), δ(CH_2_)], 1246 [ω(CH_2_), δ(COH)], 1176, 1030, 835, 683, 609, 534.

3-(2,5-Dioxo-4,4-diphenylimidazolidin-1-yl)propanoic acid, **HL3**: White solid; **^1^H NMR** (500 MHz, DMSO-*d_6_*) *δ* 2.47 (t, 2H, ^3^*J*_(H,H)_ = 7.5 Hz, C^2^H), 3.56 (t, 2H, ^3^*J*_(H,H*)*_ = 7.5 Hz, C^3^H), 7.24–7.46 (m, 10H, C^8^H–C^18^H), 9.60 (s, 1H, NH), 12.36 (s, 1H, COOH) ppm.^**13**^**C NMR** (125 MHz, DMSO-*d_6_*) *δ* 32.6 (C^2^), 35.3 (C^3^), 70.0 (C^6^), 127.6 (C^8^, C^12^, C^14^, C^18^), 128.7 (C^10^, C^16^), 129.1 (C^9^, C^11^, C^15^, C^17^), 129.6 (C^7^, C^13^), 155.7 (C^5^), 172.6 (C^4^), 173.6 (C^1^), ppm. Selected **FT-IR** data (KBr) cm^−1^: 3232 [*ν*(N−H)], 3064 [*ν*(CH)], 2945 [*ν*(CH)], 2635 [*ν*(CH2)], 1777 [*ν*(C=O), hydantoin ring], 1701 [*ν*(C=O), carboxyl group], 1451 [*ν*(C=C)], 1419 [*δ*(CO), *δ*(CH_2_)], 1227 [*ω*(CH_2_), *δ*(COH)], 1141, 946, 696, 598.

### 3.3. Synthesis of Organotin(IV) Compounds

Triphenyltin(IV) compounds were synthesised as follows: a suspension of the ligand precursors, 0.5 mmol of HL1 (147 mg), HL2 (177 mg), and HL3 (162 mg), in 5 mL of distilled water was treated with a solution of 1 M LiOH (0.5 mL, 0.5 mmol) and a clear solution formed after 3 h of stirring at 40 °C. Then 5 mL of a methanolic solution of Ph_3_SnCl (0.193 g, 0.5 mmol) was added. The mixture was stirred for 6 h while a white precipitate formed. The solvent was then filtered off, and the precipitate (product) washed with 3 mL of distilled cold water and dried in *vacuum* over silica gel.

(3-(4,5-Diphenyloxazol-2-yl)propanoato)triphenyltin(IV), **Ph_3_SnL1**: White solid; Yield: 58%. Anal. calcd. for C_36_H_29_NO_3_Sn: C, 67.32; H, 4.55; N, 2.18%. Found: C, 67.28; H, 4.60; N, 2.15%. ^1^H NMR (500 MHz, CDCl_3_): *δ* 3.00 (t, 2H, ^3^*J*_(H,H*)*_ = 7.5 Hz, C^2^H), 3.20 (t, 2H, ^3^*J*_(H,H)_ = 7.5 Hz, C^3^H), 7.27–7.35 (m, 6H, C^8^H, C^9^H, C^10^H, C^15^H, C^16^H, C^17^H), 7.35–7.48 (m, 9H, C^3′^H, C^4′^H, C^5′^H), 7.49–7.61 (m, 4H, C^7^H, C^11^H, C^14^H, C^18^H), 7.60–7.80 (m, 6H, C^2′^H, C^6′^H) ppm. ^13^C NMR (125 MHz, CDCl_3_): *δ* 24.6 (C^3^), 31.1 (C^2^), 126.6 (C^7^, C^11^, C^14^, C^18^), 127.9 (C^4′^), 128.4 ( C^8^, C^10^, C^15^, C^17^), 128.9 (C^3′^, C^5′^), 130.0 (C^9^,C^16^), 132.8 (C^13^), 134.6 (C^12^), 135.1 (C^6^), 136.8 (C^2′^, C^6′^), 138.0 (C^1′^), 145.2 (C^5^), 162.1 (C^4^), 178.3 (C^1^) ppm. ^119^Sn NMR (149.2 MHz, CDCl_3_): *δ* −96.1 ppm. Selected FT-IR data (ATR) cm^−1^: 3049 [*ν*(CH)], 2930 [*ν*(CH)], 2611 [*ν*(CH2)], 1648 [*ν*(C=O)], 1575 [*ν*(CC)], 1537 [*ν*(CC)], 1432 [*δ*(CO), *δ*(CH_2_)], 1400 [*ω*(CH), *ν*(CC)], 1313 [*ω*(CH_2_), *δ*(COH)], 1076 [*δ*(CCC), *ω*(CH)], 965 [*τ*(HCCH)], 732 [*δ*(CH2), *γ*(COC)], 454 [*ν*(Sn–O)]. ESI-MS (MeCN): *m*/*z* 643.32 [M + H]^+^.

(3-(4,5-*bis*(4-Methoxylphenyl)oxazol-2-yl)propanoato)triphenyltin(IV), **Ph_3_SnL2**: White solid; Yield: 60%. Anal. calcd. for C_38_H_33_NO_5_Sn: C, 64.98; H, 4.74; N, 1.99%. Found: C, 64.88; H, 4.79; N, 2.01%. ^1^H NMR (500 MHz, CDCl_3_): *δ* 2.92 (t, 2H, ^3^*J*_(H,H)_ = 7.5 Hz, C^2^*H*), 3.21 (t, 2H, ^3^*J*_(H,H)_ = 7.5 Hz, C^3^H), 3.83 (s, 3H, CH_3_O–Ph), 3.87 (s, 3H, CH_3_O–Ph), 6.60 – 7.10 (m, 4H, C^8^H, C^10^H, C^15^H, C^17^H), 7.30–7.50 (m, 9H, C^3′^H, C^4′^H, C^5′^H), 7.69–7.90 (m, 10H, C^7^H, C^11^H, C^14^H, C^18^H, C^2′^H, C^6′^H) ppm. ^13^C NMR (125 MHz, CDCl_3_): δ 25.9 (C^3^), 32.9 (C^2^), 55.5 (CH_3_O–Ph), 114.0 (C^8^, C^10^), 127.0 (C^14^, C^18^), 128.0 (C^4′^), 128.3 (C^2′^, C^6′^), 128.4 (C^13^), 129.1 (C^7^, C^11^), 132.6 (C^6^), 135.2 (C^5^), 135.9 (C^3′^, C^5′^), 137.5 (C^1′^), 145.3 (C^12^), 157.1 (C^4^), 160.4 (C^9^, C^16^), 169,1 (C^1^), ppm. ^119^Sn NMR (149.2 MHz, CDCl_3_): *δ* −95.0 ppm. Selected FT-IR data (ATR) cm^−1^: 3050 [*ν*(CH)], 2969 [ν(CH)], 2928 [ν(CH2)], 2836 [ν(CH2)], 1713 [ν(C=O)], 1591 [ν(CC)], 1514[ν(CC)], 1452 [δ(CO), δ(CH_2_)], 1242 [ω(CH_2_), δ(COH)], 1169, 1034, 840, 683, 611, 529, 447 [*ν*(Sn–O)]. ESI-MS (MeCN): *m*/*z* 703.41 [M + H]^+^.

(3-(2,5-Dioxo-4,4-diphenylimidazolidin-1-yl)propanoato)triphenyltin(IV), **Ph_3_SnL3**: White solid; Yield: 65%. Anal. calcd. for C_36_H_30_N_2_O_4_Sn: C, 64.21; H, 4.49; N, 4.16%. Found: C, 64.31; H, 4.42; N, 4.07%. ^1^H NMR (500 MHz, CDCl_3_): *δ* 2.79 (t, 2H, ^3^*J_(H,H)_* = 7.5 Hz, C^2^H), 3.89 (t, 2H, ^3^*J_(H,H)_* = 7.5 Hz, C^3^H), 7.10–7.27 (m, 10H, C^8^H–C^18^H), 7.30 (s, 3H, C^4′^H), 7.41 (s, 6H, C^3′^H, C^5′^H), 7.68 (s, 6H, C^2′^H, C^6′^H) ppm. ^13^C NMR (125 MHz, CDCl_3_) *δ* 32.5 (C^2^), 38.1 (C^3^), 70.1 (C^6^), 126.5 (C^8^, C^12^, C^14^, C^18^), 128.3 (C^2′^, C^6′^), 129.0 (C^9^, C^11^, C^15^, C^17^), 129.4 (C^10^, C^16^), 131.9 (C^4′^), 137.0 (C^3′^, C^5′^), 138.1 (C^7^, C^13^), 140.0 (C^1′^), 156.2 (C^5^), 173.2 (C^4^), 173.7 (C^1^) ppm. ^119^Sn NMR (149.2 MHz, CDCl_3_): *δ* −89.3 ppm. Selected FT-IR data (ATR) cm^−1^: 3500 [*ν*(NH)], 3051 [*ν*(CH)], 2940 [*ν*(CH)], 1770 [*ν*(C=O), hydantoin ring], 1710 [*ν*(C=O), carboxyl group], 1430 [*ν*(C=C)], 1419 [*δ*(CO), *δ*(CH_2_)], 1225 [*ω*(CH_2_), *δ*(COH)], 1140, 946, 696, 598, 449 [*ν*(Sn–O)]. ESI-MS (MeCN): *m*/*z* 674.38 [M + H]^+^.

### 3.4. Cell Lines, General Conditions and IC_50_ Determination

The cell proliferations of ligand precursors and organotin(IV) compounds were evaluated with the aid of MTT (3-(4,5-dimethylthiazol-2-yl)-2,5-12 diphenyltetrazolium bromide) and CV (crystal violet) assays toward prostate (PC-3), colon (HT-29), breast (MCF-7) and hepatic (HepG2) cancer cell lines, and normal mouse embryonic fibroblast cells (NIH 3T3). The cells were cultivated in T75 flasks containing 10 mL of complete medium and kept at 37 °C and 5% CO_2_ with the aim of detaching the cells. The nutrient medium was RPMI-1640 (PAA Laboratories) supplemented with 10% foetal bovine serum, L-glutamine, and penicillin/streptomycin (Sigma). Afterward, 5.000 cells of PC-3 and MCF-7 and 1.500 cells of HT-29 and HepG2 were transferred into 96 well-plates and used for MTT and CV assays. For microscopy investigations as well as FACS measurements (except for ROS/RNS measurements), MCF-7 cells were seeded overnight in 6-well plates (150,000 cells/well) in 10 mL of complete medium. After 24 h, cells were treated with **Ph_3_SnL1** or cisplatin for 24 and 48 h in the concentrations corresponding to their IC_50_ and 2 × IC_50_ values and used in various assays as indicated below. In all assays, control cells were untreated MCF-7 cells incubated for the same time period as treated cells and treated with appropriate reagents in various assays. BD FACS Aria III was used for flow cytometry experiments and results were analysed using BD FACSDiva software (BD Biosciences, Heidelberg, Germany) and Cyflogic (V 1.2.1, CyFlo Ltd., Turku, Finland). A fluorescence microscope Invitrogen EVOS FL 2 Auto 2 Cell Imaging System was used for visualization of the cells as well as for detection of fluorescence in the cells.

Ligands **HL1**‒**HL3**, organotin(IV) compounds, and cisplatin were investigated in 9 different concentrations (100, 50, 10, 5, 1, 0.5, 0.1, 0.05, 0.01 μM). Using the complete medium, each stock solution of the test compounds (20 mM in DMSO) was diluted to obtain the working solution. The final concentration of DMSO never overstepped 0.5%, which is non-toxic to the cells [37]. Three technical replicates were performed for each concentration. The treated cells were incubated for 48 h at 37 °C under 5% CO_2_ and the viability of the cells was determined using MTT and CV assays according to the literature procedure [28,38,39]. The IC_50_ values, defined as 50% inhibitory concentration, were calculated with a four-parameter logistic function and presented in a mean. All experiments were performed in triplicate.

### 3.5. Metal Uptake

The tin and platinum uptake analysis was performed on MCF-7 cells with the aid of inductively coupled plasma mass spectrometry (ICP-MS). The cells were seeded in T25 flasks and allowed to grow. After 24 h, the exponentially growing cells were treated with 10 mL of **Ph_3_SnL1** and cisplatin in concentrations corresponding to their IC_50_ values for 24 h, and thereafter trypsinised and collected by centrifugation at 1200 rpm for 8 min. Using 5 mL of ice-cold PBS the cells were washed and the metal uptake concentration in prepared samples was determined according to the previously described procedure [40]. A Thermo Scientific iCAP Qc ICP-MS (Thermo Scientific, Bremen, Germany) instrument using the supplied autotune protocols and operational software Qtegra was employed for metal uptake determination. The measurements were performed on isotopes ^119^Sn, ^120^Sn, ^195^Pt, and ^198^Pt and the concentrations of metals were expressed as ppb (µg/L).

### 3.6. Morphological Analysis (AO and DAPI Staining)

The morphological changes of MCF-7 cell death induced by **Ph_3_SnL1** and cisplatin were studied by AO (acridine orange) [41] and DAPI (4′,6-diamidino-2-phenylindole) [42] staining, and examined under a fluorescence microscope. Acridine orange can cross the cell membrane and viable and early apoptotic cells can be detected. Chromatin condensation, seen as dense green areas, or membrane blebbing, both appearing in apoptosis, is easily proven by AO staining [40]. After 24 h of treatment (IC_50_ and 2 × IC_50_ concentrations) the cells were stained with acridine orange (3 μg/mL AO in PBS), and observed with a fluorescence microscope (Fluorescence microscope-Invitrogen EVOS FL 2 Auto 2 Cell Imaging System). On the other hand, for the DAPI assay 0.5 mL of Triton (0.1% in PBS) was added to the treated cells after the fixing the cells with 0.5 mL of 4% paraformaldehyde (PFA) for 8 min. Thereafter, 0.5 mL of DAPI solution was used to stain the cells for 5 min in the dark [42]. The morphological characteristics were visualized under a fluorescence microscope (Fluorescence microscope-Invitrogen EVOS FL 2 Auto 2 Cell Imaging System).

### 3.7. Activation of Caspases

After 48 h of incubation (with IC_50_ concentrations) the apostat assay was performed as indicated by the supplier. The cells were resuspended in 1 mL of PBS and analysed with flow cytometry [27].

### 3.8. Autophagy Analysis

The MCF cells were incubated at IC_50_ concentrations of **Ph_3_SnL1** and cisplatin for 48 h at 37 °C under 5% CO_2_. Thereafter, the medium was collected into 15 mL Falcon tubes, while the cells were washed with 1 mL of PBS. The cells were detached using 0.5 μL of trypsin. The deactivation of trypsination was performed with the formerly collected medium, while for the untreated cells, 1 mL of complete medium was added, and the cells were collected by the pipette tip harvesting technique from each well into a new Falcon tube, which then was centrifuged for 3 min at 1000 rpm. The supernatant was discarded, and the cells were washed with 1 mL of PBS, centrifuged for 3 min at 1000 rpm and then the supernatant was discarded. Afterwards, 500 μL of the acridine orange (AO) working solution was used to stain the cells which then were incubated for 30 min at 37 °C under 5% CO_2_. The deactivation of the staining process was performed by adding 1 mL of PBS, and after centrifugation for 3 min at 1000 rpm, the supernatant was discarded. The cells were resuspended in 1 mL of PBS and analysed by flow cytometry [27].

### 3.9. Cell Cycle Analysis

Upon 48 h of treatment at IC_50_ concentrations, the MCF-7 cells were detached using 0.5 μL of trypsin. The Falcon tubes were centrifuged for 3 min at 1000 rpm and the supernatant was discarded. Afterwards, the cells were washed with 1 mL of PBS, centrifuged for 3 min at 1000 rpm and the supernatant was discarded. The cells were resuspended in 300 μL of ice-cold PBS buffer and then fixed by adding them dropwise to 700 μL of ice-cold absolute ethanol. The cells were stored for 24 h at 4 °C, then centrifuged for 3 min at 1000 rpm, washed with PBS and stained with 1 mL of DAPI (4′,6-diamidino-2-phenylindole) working solution [39]. The prepared cells were incubated at room temperature for 10 min and analysed by flow cytometry.

### 3.10. Investigation of ROS/RNS Production

For the investigation of ROS/RNS production the cells from the T25 flask were collected by applying 1 mL of trypsin and diluted with 9 mL of complete medium. Using 1 mL of DHR (dihydrorhodamine) working solution the cells were stained and incubated at 37 °C and 5% CO_2_ for 10 min. The staining process was stopped by applying 4 mL of complete medium, then the solution was centrifuged for 3 min at 1000 rpm and the supernatant was discarded. The cells were resuspended in complete medium (1 mL/well) and seeded in a 6 well plate. After 24 h incubation, the cells were treated with IC_50_ of the tested compounds and incubated for 48 h at 37 °C and 5% CO_2_. The cells were detached using 0.5 μL of trypsin and incubated for 3 min at 37 °C and 5% CO_2_. The trypsinisation was stopped by adding 1 mL of complete medium to each well, and cells were centrifuged at 1000 rpm for 3 min, washed with 1 mL of PBS, centrifuged, and then resuspended in 1 mL of PBS. The prepared cells were analysed by flow cytometry [27].

### 3.11. Investigation of NO Production

For NO (nitric oxide) production evaluation, cells were treated at IC_50_ of the test compounds and incubated for 48 h at 37 °C under 5% CO_2_. The cells were washed with PBS and then treated with 1 mL of DAF-FM (4-amino-5-methylamino-2,7′-diflurofluorescin diacetate) working solution. The treated cells were incubated for 1 h at 37 °C and 5% CO_2_. The stain was deactivated by incubation in a medium for 15 min before the cells were washed with 1 mL of PBS. The cells were detached using 0.5 μL of trypsin and incubated for 3 min at 37 °C and 5% CO_2_, centrifuged at 1000 rpm for 3 min, and washed with 1 mL of PBS. The prepared cells were analysed by flow cytometry [39].

## 4. Conclusions

Three novel triphenyltin(IV) compounds based on commercial drugs and their derivatives have been synthesised, characterised and tested in vitro for antiproliferative activity on four different tumour cell lines. The ligand precursor oxaprozin (3-(4,5-diphenyloxazol-2-yl)propanoic acid), **HL1**, and the new propanoic acids derivatives 3-(4,5-bis(4-methoxylphenyl)oxazol-2-yl)propanoic acid, **HL2**, and 3-(2,5-dioxo-4,4-diphenylimidazolidin-1-yl)propanoic acid, **HL3**, did not affect cell proliferation in any cancer cell line tested. Conversely, the IC_50_ values of the triphenyltin(IV) compounds ranged from 0.100 to 0.758 µM. Complex **Ph_3_SnL1** emerged as the most potent cytotoxic agent against MCF-7 cells (CV assay: IC_50_ = 0.218 ± 0.025 µM). Different biological assays for the investigation of its mechanism of action suggest that **Ph_3_SnL1** induces caspase-independent apoptosis in MCF-7 cells. Moreover, ICP-MS analysis indicates that this is achieved with lower intracellular concentrations of tin in MCF-7 cells than platinum. Considering the required extracellular concentrations, the triphenyltin(IV) compounds show a better cell bioavailability and accumulation, a common phenomenon for lipophilic (and cationic) substances. In conclusion, **Ph_3_SnL1** emerges as a novel and promising potential antiproliferative drug which should be confirmed by a series of novel experiments. Although neither ROS nor RNS—often related to the cell’s energy metabolism [43]—are involved, subcellular investigation and the molecular mode of action of triphenyltin(IV) compounds remain to be studied in biochemical follow-up research.

## Data Availability

Data supporting obtained results can be obtained from the authors upon request.

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
