# Peer review of "In Vitro Evaluation of Antiproliferative Properties of Novel Organotin(IV) Carboxylate Compounds with Propanoic Acid Derivatives on a Panel of Human Cancer Cell Lines"

_molecules, 2021, doi:10.3390/molecules26113199_

Round 1

Reviewer 1 Report

The manuscript of Pantelic et al entitled "In vitro evaluation of antiproliferative properties of novel or3 ganotin(IV) carboxylate compounds with propanoic acid deriv4 atives on a panel of human cancer cell lines" presents very interesting results regarding a compound that appears to be more effective than cis-platin in treating breast cancer. 

I have few questions and suggestions for the authors:

  1. The authors say that: In all assays, control
    366 cells are untreated MCF-7 cells incubated for the same time period as treated cells. Is it true? If it is true, then the results reported in Sections 2.4 and 2.5 are completely unexplained. Figures 4 and 5 are not correctly interpreted. I recommend the authors to reconsider their interpretation. 
  2. The authors do not have enough arguments to confirm their conclusions that compound 1 is more efficient than cis-platin. I suggest to them to make supplementary experiments for MTS viability tests, intracellular ROS production, and cellular DNA fragmentation. It is mandatory to perform these tests in order to explain their results and to sustain their conclusions. Here is an article which includes the above-mentioned tests Popescu, T., Matei, C.O., Vlaicu, I.D. et al. Influence of surfactant-tailored Mn-doped ZnO nanoparticles on ROS production and DNA damage induced in murine fibroblast cells. Sci Rep 10, 18062 (2020). https://doi.org/10.1038/s41598-020-74816-0. 
  3. Please reconsider all your results and conclusions after correlating them with the results from the tests at point 2.

Author Response

We thank the reviewers for constructive suggestions. All corrections in the manuscript were made using the Track Changes option and highlighted yellow. Bellow, we are giving a point-by-point description of how the suggested corrections are carried out.

Point 1: The authors say that: In all assays, control cells are untreated MCF-7 cells incubated for the same time period as treated cells. Is it true? If it is true, then the results reported in Sections 2.4 and 2.5 are completely unexplained. Figures 4 and 5 are not correctly interpreted. I recommend the authors to reconsider their interpretation. 

Response 1: Thank you for the careful evaluation of our manuscript. Untreated cells were stained with appropriate reagents in different flow cytometric assays (we clarified this now in line 393). We reevaluated findings in Figures 4 and 5 and didn’t find any misleading interpretation.

Point 2: The authors do not have enough arguments to confirm their conclusions that compound 1 is more efficient than cis-platin. I suggest to them to make supplementary experiments for MTS viability tests, intracellular ROS production, and cellular DNA fragmentation. It is mandatory to perform these tests in order to explain their results and to sustain their conclusions. Here is an article which includes the above-mentioned tests Popescu, T., Matei, C.O., Vlaicu, I.D. et al. Influence of surfactant-tailored Mn-doped ZnO nanoparticles on ROS production and DNA damage induced in murine fibroblast cells. Sci Rep 10, 18062 (2020). https://doi.org/10.1038/s41598-020-74816-0.

Response 2: Thank you for your suggestion. However, it is clearly seen by comparison of IC50 values that compound 1 is more efficient than cisplatin. Moreover, the intracellular ROS production is already reported in the manuscript (see lines: 271-283). As we showed with two assays that the condensation of nucleus upon treatment occurs, the additional third assay (DNA laddering) would not give any additional information.

Point 3: Please reconsider all your results and conclusions after correlating them with the results from the tests at point 2.

Response 3: As suggested, we reconsidered all results reported in this manuscript; however additional experiment concerning DNA laddering was not performed.

Reviewer 2 Report

In this paper, Pantelić et al. described synthesis, chemical characterization and biological features of three novel organotin compounds. The topic is undoubtely of high importance for the bioinorganic chemists community and the presented results could be meritable of publication. Anyway, in the current version of the manuscript, there are many relevant points that must be improved before the acceptance of the work:

-First of all the authors must improve the quality of the text, since some sentences are poorly written (for example at rows 73-76; 82-84; 158-160).

-The characterization of the investigated tin compounds is satisfying, anyway 119SnNMR experiments should be performed. These experiments will be useful to the readers and moreover will give to authors a stronger information about the complexes stability (much more than 13CNMR experiments).

-The authors must determine the IC50 of the investigated compounds at least on a healthy cell line, sinche without a clear evidence of a differential effect is not possible to assess either a vague importance of these compounds. Moreover, since the high toxicity of many organotin derivatives, this point should be considered as mandatory.

-In conclusive paragraph the authors claim mitochondria as possible targets of the investigated compounds. At this point of research, this statement sounds quite speculative. Sinche a cell isn't made only of DNA and mitochondria, the authors shlould provide at least a preliminary evidence about as a proof for this sentence (for example with a commercially availlable thioredoxin reductase inhibition test). If not, the authors should strongly redimensionate that statement.

Minor points:

Row 66: remove "chloride"

Please replace 1, 2 and 3 with Ph3SnL1, Ph3SnL2 and Ph3SnL3 in the whole manuscript (text, captions, schemes and so on). The numeric notation turned out to be quite annoying since it is the abbreviation of the abbreviated name.

Paragraph 2.2: Some redundant sentences can be found. Correct them

Author Response

We thank the reviewers for constructive suggestions. All corrections in the manuscript were made using the Track Changes option and highlighted yellow. Bellow, we are giving a point-by-point description of how the suggested corrections are carried out.

Point 1: First of all the authors must improve the quality of the text, since some sentences are poorly written (for example at rows 73-76; 82-84; 158-160).

Response 1: Thank you for your observation. The English language of the manuscript has been improved in its current form.

Point 2: The characterization of the investigated tin compounds is satisfying, anyway 119Sn NMR experiments should be performed. These experiments will be useful to the readers and moreover will give to authors a stronger information about the complexes stability (much more than 13CNMR experiments).

Response 2: Thank you for your comment. The 119Sn NMR spectra of synthesized compounds are included and discussed in the manuscript. We agree with Reviewer #2, though due to the actual pandemic situation (restriction in the work of operators and large waiting queue) we were able to record only a single 119Sn NMR spectrum per compound. We do hope that time-dependent 13C NMR is sufficient to prove the stability of the organotin(IV) compounds.

Point 3: The authors must determine the IC50 of the investigated compounds at least on a healthy cell line, since without a clear evidence of a differential effect is not possible to assess either a vague importance of these compounds. Moreover, since the high toxicity of many organotin derivatives, this point should be considered as mandatory.

Response 3: As suggested by Reviewer #2. additional experiments were performed.  The activity of compounds toward normal cells (mouse embryotic fibroblast - NIH3T3) was determined and appropriate discussion is included in the revised version of the manuscript.

Point 4: In conclusive paragraph the authors claim mitochondria as possible targets of the investigated compounds. At this point of research, this statement sounds quite speculative. Since a cell isn't made only of DNA and mitochondria, the authors shlould provide at least a preliminary evidence about as a proof for this sentence (for example with a commercially availlable thioredoxin reductase inhibition test). If not, the authors should strongly redimensionate that statement.

Response 4: We agree with Reviewer #2, the conclusion statement is reformulated.

Point 5: Row 66: remove "chloride"

Response 5: We are sorry for this typing mistake. The “chloride” is removed.

Point 6: Please replace 1, 2 and 3 with Ph3SnL1, Ph3SnL2 and Ph3SnL3 in the whole manuscript (text, captions, schemes and so on). The numeric notation turned out to be quite annoying since it is the abbreviation of the abbreviated name.

Response 6: As you suggested, the numeric notation 1, 2, and 3 are replaced with Ph3SnL1, Ph3SnL2 and Ph3SnL3 in the whole manuscript.

Point 7: Paragraph 2.2: Some redundant sentences can be found. Correct them

Response 7: Thank you for your observation. Some words in the sentences chanced places during the packing of the text in the manuscript template. The sentences are corrected in the current form of the manuscript.

Round 2

Reviewer 1 Report

Dear authors,

I believe you managed to answer to almost all of my questions and recommendations. However, if you compare the toxicity of your complexes with cisplatin toxicity against healthy cells, which would you rather recommend as anticancer agents? Did you consider this comparison? I understand that you say that your compound presents selectivity, but even so what would you recommend? 

Best regards! 

Reviewer 2 Report

Since authors clarified all highlighted criticism I recommand the acceptance of the present version of this study